# Economic burden of maternal morbidity – A systematic review of cost-of-illness studies

Patrick S. Moran[1]*, Francesca Wuytack[1], Michael Turner[2], Charles Normand[3,4], Stephanie Brown[5,6], Cecily Begley[1], Deirdre Daly[1]

**1** School of Nursing and Midwifery, Trinity College, Dublin, Ireland, **2** School Of Medicine, University College, Dublin, Ireland, **3** Centre for Health Policy and Management, Trinity College, Dublin, Ireland, **4** Cicely Saunders Institute, King's College, London, United Kingdom, **5** General Practice and Primary Health Care Academic Centre, University of Melbourne, Melbourne, Australia, **6** Murdoch Children's Research Institute, Melbourne, Australia

* moranp6@tcd.ie

**Data Availability Statement:** All data is available from the cited sources and in the supplementary material published with this paper.

## Abstract

### Aim

To estimate the economic burden of common health problems associated with pregnancy and childbirth, such as incontinence, mental health problems, or gestational diabetes, excluding acute complications of labour or birth, or severe acute adverse maternal outcomes.

### Methods

Searches for relevant studies were carried out to November 2019 in Medline, Embase, CINAHL, PsycINFO and EconLit databases. After initial screening, all results were reviewed for inclusion by two authors. An adapted version of a previously developed checklist for cost-of-illness studies was used for quality appraisal. All costs were converted to 2018 Euro using national consumer price indices and purchasing power parity conversion factors.

### Results

Thirty-eight relevant studies were identified, some of which reported incremental costs for more than one health problem (16 gestational diabetes, 13 overweight/obesity, 8 mental health, 4 hypertensive disorders, 2 nausea and vomiting, 2 epilepsy, 1 intimate partner violence). A high level of heterogeneity was observed in both the methods used, and the incremental cost estimates obtained for each morbidity. Average incremental costs tended to be higher in studies that modelled a hypothetical cohort of women using data from a range of sources (compared to analyses of primary data), and in studies set in the United States. No studies that examined the economic burden of some common pregnancy-related morbidities, such as incontinence, pelvic girdle pain, or sexual health problems, were identified.

### Conclusion

Our findings indicate that maternal morbidity is associated with significant costs to health systems and society, but large gaps remain in the evidence base for the economic burden of

**Funding:** This research was funded by the Health Research Board in Ireland (grant ref: HRB ICE-2015-1019). The funders had no role in study design, data collection and analysis, decision to publish, or preparation of the manuscript.

**Competing interests:** The authors have declared that no competing interests exist.

some common health problems associated with pregnancy and childbirth. More research is needed to examine the economic burden of a range of common maternal health problems, and future research should adopt consistent methodological approaches to ensure comparability of results.

## Introduction

Pregnancy and childbirth can be a significant cause of morbidity in women. Globally, the incidence of maternal disorders was estimated to be almost 80 million cases in 2017, corresponding to over 800,000 years lived with disability (YLDs).[1] As high as these figures are, they underestimate the true extent of the problem because they are focused on obstetric complications during labour and birth, such as haemorrhage, sepsis, hypertensive disorders and obstructed labour. The reported incidence does not include many other health problems that are common during pregnancy and postpartum, such as depression, incontinence, sexual health issues, and pelvic girdle pain.[2–6] These types of health problems are frequently underreported and undertreated, due to their sensitive nature, or a belief that they are normal, self-limiting symptoms of pregnancy and birth.[7–9] There is a growing body of research describing their prevalence, which indicates that almost all women (94%) experience at least one major health problem in the first year after having a baby, with up to one in five (20%) reporting depressive symptoms and almost half (47%) reporting urinary incontinence.[9–11] Postnatal morbidity can also be influenced by mode of birth or any complications during or immediately after the birth, such as postpartum haemorrhage.[12–14] While the magnitude of the clinical burden associated with these issues is becoming increasingly clear, there is a lack of data on the economic burden that they impose on women, families, and the health system.

Cost of illness (COI) studies are designed to measure all the direct and indirect costs associated with a disease or diseases, to provide an estimate of the total burden that these conditions impose on society. This estimate represents the total economic value of the additional resources that are currently required by those who are effectively diagnosed and treated, as well as the costs of dealing with the consequences for those who are not optimally managed. COI studies were among the earliest forms of economic evaluation to be carried out in healthcare, and the strengths and weaknesses of this approach have been extensively debated.[15, 16] Among the benefits of COI analysis is that it provides a detailed description of how much society is spending on a particular health problem, and therefore how much would be saved if it could be eradicated completely. Knowing the relative scale of resource use associated with different health problems may help policymakers prioritise areas for improvement, by highlighting where the greatest savings can be made. The main problem with COI studies from an economic perspective is that they do not link costs to health outcomes, so they do not provide information that can be used to guide decisions about efficient resource allocation. As has been pointed out, just because a particular disease is associated with a high cost to society does not mean it should be prioritised for funding, since the reason the costs are so high might be that it is well funded already.[17] In addition, the COI estimate provides no indication of how eradicable a given disease may be, or whether improving diagnosis and treatment would cost more or less than the status quo. It is worth noting that these limitations apply equally to prioritisation based on the clinical burden of disease, since knowing which conditions account for the most deaths does not take into account how preventable those deaths are.

While there are clearly limitations to what one can achieve through an analysis of the economic burden of disease, these studies can offer some very useful information provided that

careful scrutiny is given to the context that pertains within a given clinical area. This includes assessment of the extent to which case detection and clinical management are already considered optimal, as well as distinguishing between the costs of clinical intervention and the costs of allowing diseases to take their natural course. Interpreted correctly, COI studies can provide valuable descriptive information on the individual cost components associated with maternal health problems, the level of variability that surrounds them, and provide insights into the range and behaviour of each of the relevant cost components to aid the design of future economic evaluations.

The aim of this systematic review is to identify and synthesise the existing evidence on the economic burden of common health problems women experience over the course of their pregnancy and postpartum, excluding acute complications of labour or birth, or severe acute adverse maternal outcomes.

## Methods

We searched for studies reporting the incremental costs associated with health problems experienced by women during pregnancy and postpartum. The population of interest were women who experienced morbidity related to pregnancy and childbirth, before or after the birth, such as incontinence, mental health problems, gestational diabetes, obesity or hypertension. We excluded studies that examined acute complications of labour or birth (for example haemorrhage, uterine rupture, or sepsis) or severe acute adverse maternal outcomes (for example thromboembolism, eclampsia or acute renal failure).

Eligible studies were those that reported incremental costs associated with a relevant condition, which is the additional costs over and above those that would be incurred in the absence of that condition. Modelling studies estimating the economic burden of disease using data from a range of different sources were also included. Studies that were published as conference abstracts were excluded. Studies that reported incremental costs associated with a particular intervention in treatment versus control groups of women with a given morbidity were excluded on the basis that they did not provide data on the incremental costs of the condition itself. Also excluded were studies that only reported the average costs of care for women with a particular health problem without having a comparison group of women without that problem. Studies reporting costs associated with multiple pregnancies, birth defects, assisted reproduction, preterm birth, substance abuse, or alternative models of maternity care were also excluded.

The primary outcome of interest was the incremental cost of maternal health problems, expressed as the absolute cost difference (in 2018 Euro) of treating women with and without these maternal health problems. Costs estimated from the perspective of the health service (payer perspective) were included along with those estimated from the broader societal perspective, which includes direct or indirect costs that fall outside the health service. The accrual period over which costs were calculated was reported for all studies.

Searches for relevant studies were carried out between November 2017 and November 2019 in Medline, Embase, CINAHL, PsycINFO and EconLit citation databases. A two-stage search strategy was used, which involved an initial broad search with high expected sensitivity, followed by a more targeted search designed for greater specificity (see S1 Table for details). No date or language limits were applied. This review is one strand of a review protocol published on the Prospero database (CRD42017077722).

Results were initially screened by one reviewer to exclude irrelevant studies based on title and abstract. All remaining studies were independently reviewed by two people. Conflicts in regard to inclusion and exclusion of studies were resolved through discussion. Quality appraisal of each study was carried out using an adapted version of a previously developed critical

appraisal checklist for COI studies.[18] This involved appraising each potentially relevant study for relevance, methodological rigour, and reporting. Studies that failed to meet the minimum requirement at any stage were excluded without proceeding to the next stage. Therefore all included studies were those that met an acceptable level across all three appraisal stages. Data extraction from relevant studies was carried out by one reviewer using a predefined data extraction template and checked by another reviewer for accuracy and completeness. Reporting of the review was done in accordance with PRISMA guidelines (see S1 Checklist).

All costs reported in included studies were converted to 2018 Euro by first inflating the base currency and then converting to Euro using the Organisation for Economic Co-operation and Development (OECD) purchasing power parity index for the 19 Euro area countries.[19] All incremental costs are reported as absolute money amounts, as well as proportional differences. Given the challenges in pooling cost data from different regions or derived using different methodological approaches, a narrative synthesis of the economic burden of maternal morbidity was performed within each disease area. Where necessary, estimates from subgroups within individual studies were combined to produce a weighted mean incremental cost for a larger subgroup that allowed for comparisons across studies. For example, this included combining data for women classified using different obesity levels (I, II, III) into one category (obese, body mass index [BMI] ≥30). While this was necessary to facilitate comparisons with other studies that used similar groupings, a limitaton of this is that it does not permit comparisons across different obesity categories (e.g. class I versus class III obesity).

## Results

The search identified 6,254 citations, 63 of which were reviewed in detail following screening of title and abstracts. Twenty-five articles were excluded based on full-text review, leaving a total of 38 included studies (Fig 1). These studies were all published between 2001 and 2019, with the majority set either in the United States (US) (15 studies, 39%) or the United Kingdom (UK) (8 studies, 22%).

Diabetes was the most common maternal morbidity examined, with a total of 16 studies reporting incremental costs associated with gestational diabetes, with some also reporting costs associated with a prior diagnosis of type I or II diabetes mellitus. Thirteen studies reported costs associated with overweight or obesity in pregnancy, eight reported costs related to antenatal or postnatal mental health problems, and the remaining studies reported incremental cost estimates for hypertensive disorders (4 studies), nausea and vomiting (2 studies), epilepsy (2 studies), and intimate partner violence (1 study). The majority of studies (73%) were analyses of cross-sectional data, two studies involved longitudinal analysis of costs over time, and eight were modelling studies that estimated incremental costs by combining data from multiple sources. Complete data extraction tables are provided in the S2 Table.

In general, there was substantial methodological heterogeneity both in the perspective and the time periods used to calculate costs. Most studies (31, 82%) adopted a payer perspective that limited the analysis of costs to those that fell on the health system or insurer, omitting direct costs that were covered by out-of-pocket payments (OOP) by women themselves, or productivity losses due to time off work. Seven studies adopted a broader societal perspective that included costs to women along with costs to the health service, but only five of these included productivity losses.[20–24]

### Quality appraisal

Quality appraisal was designed to evaluate studies based on an assessment of relevance, methodological rigour, and reporting. Rather than providing an index score for each study, the tool

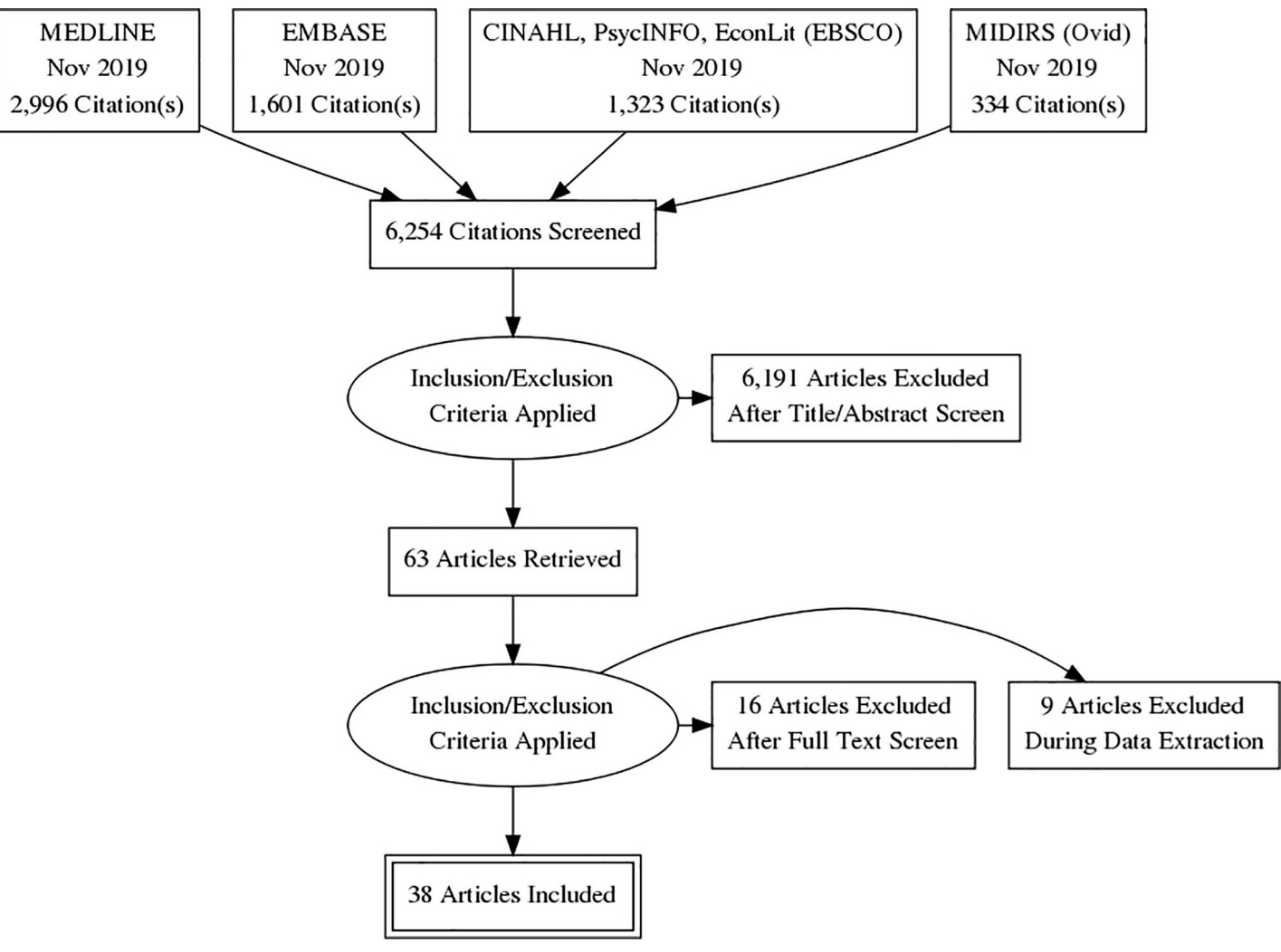

**Fig 1. PRISMA flowchart of search results.**

provided a checklist to highlight potential weaknesses in included studies in each of these areas. Results of the quality appraisal are shown in S3 Table. From a methodological perspective, the two areas of greatest concern were the omission of costs that fall outside the health service (82% of studies), and the reporting of uncertainty associated with the incremental cost estimates. There was also a lack of sensitivity analysis exploring the impact of uncertainty relating to the prevalence of particular health problems, treatment rates, and unit prices, with only 12 studies (3%) reporting some form of sensitivity analysis, and one reporting having used conservative estimates to mitigate parameter uncertainty. [20, 21, 25–35]

### Gestational diabetes

There was a high degree of methodological heterogeneity in the study design, cost accrual period and perspective of the 16 identified studies (Table 1). While most were analyses of cross-sectional data (10 studies), five studies modelled the increase in resource use by combining data from multiple sources, and one study involved longitudinal analysis of registry data on resource utilisation up to 14 years after the birth for women with and without gestational diabetes.

**Table 1. Incremental costs of diabetes.**

| Study (Setting) | Study design (n) | Accrual period for costs | Perspective—Costs included | Results [absolute cost increase (percentage increase)] |
|---|---|---|---|---|
| Meregaglia 2018 [33] (Italy) | Modelling (N/A) | 28 weeks gestation to birth (3 months) | Payer-Mothers and neonatal care | GDM associated with incremental costs of €839 (29%) per case |
| Xu 2017[27] (China) | Modelling (N/A) | 28 weeks gestation to birth (3 months) | Payer-Mothers and neonatal care | GDM associated with incremental costs of €1,530 (95%) per case |
| Law 2015[37] (US) | Cross-sectional (137,040) | First 3 months of life | Payer-Children only | Higher unadjusted cost for new-born care among mothers with diabetes (€1,132, 14%) |
| Law 2015[30] (US) | Cross-sectional (322,141) | During pregnancy and 3 months postpartum | Payer-Mothers only | Higher unadjusted cost of maternal care for those with diabetes (€4,000, 30%) |
| Whiteman 2015 [38] (US) | Cross-sectional (576,843) | Birth and 12 months postpartum | Payer-Mothers and children | Being overweight/obese with GDM (€1,426, 20%) was associated with higher average costs of maternal and infant care |
| Jovanovic 2015[39] (US) | Cross sectional (645,195) | Pregnancy and 3 months postpartum | Payer-Mothers and children | Significant increase in cost for all types of diabetes (T1DM: €12,561, 92%; T2DM: €7,993, 58%; GDM: €3,263, 24%, and Progressing GDM: €8,294, 61%). |
| Lenoir-Wijnkoop 2015[32] (US) | Modelling (N/A) | During pregnancy and birth | Payer-Mothers and neonatal care | Mothers with gestational diabetes (€13,680, %NR) were associated with higher costs of care. |
| Danyliv 2015[40] (Ireland) | Cross-sectional (658) | During pregnancy and annual postpartum costs | Payer-Mothers and neonatal care | GDM associated with increased cost of birth (€865, 15%) and annual care costs postpartum (€720, 133%). Equivalent to an incremental cost of €1,584 (25%) from birth to 12 months postpartum |
| Son 2014[36] (South Korea) | Cross-sectional (1,282,498) | During pregnancy and birth | Payer-Mothers only | Both GDM (€263, 11%) and pre-existing DM (€672, 27%) were associated with significantly higher costs |
| Dall 2014[23] (US) | Modelling (N/A) | During pregnancy and 12 months postpartum | Societal-Mothers and children | Excess annual costs associated with gestational diabetes were €4,893 (%NR) per woman |
| Gillespie 2013[34] (Ireland) | Cross-sectional (4,372) | During pregnancy and birth | Payer-Mothers and neonatal care | GDM associated with significantly higher total unadjusted costs of care (€2,313, 51%) |
| Cavassini 2012[41] (Brazil) | Cross-sectional (68) | During pregnancy and birth | Payer-Mothers and neonatal care | Total additional cost attributable to diabetes was €3,920 (161%) for inpatients and €185 (8%) for outpatients |
| Kolu 2012[42] (Finland) | Cross-sectional (848) | From 12 weeks gestation to discharge from hospital after birth | Societal-Mothers and neonatal care | GDM associated with significantly higher total costs of care (€1,468, 25%) |
| Anderberg 2012 [43] (Sweden) | Longitudinal (1,438) | 10 to 14 years after the birth | Payer-Mothers only | Average increase in annual costs for those with GDM at end of 14-year follow-up was €401 (39%, not statistically significant) |
| Kolu 2011[44] (Finland) | Cross-sectional (56,136) | During pregnancy only | Payer-Mothers only | All groups with GDM or GDM risk factors associated with higher mean antenatal care costs (-GDM/+risk factors: €174, 10%; +GDM/-Risk Factors: €406, 24%; +GDM/+Risk Factors: €682, 40%) |
| Chen 2009[45] (US) | Modelling (N/A) | During pregnancy and 12 months postpartum | Payer-Mothers and children | GDM associated with an incremental cost of €3,509 (%NR) per birth. |

N/A not applicable; GDM gestational diabetes mellitus; T1DM type 1 diabetes mellitus; T2DM type 2 diabetes mellitus; DM diabetes mellitus; NR not reported

All but two studies adopted a payer perspective, but within this group, there was significant variation in the costs that were included. Four studies only included costs associated with treating mothers, seven studies included costs of treating mothers as well as costs of neonatal care during the perinatal episode of care (such as intensive care unit costs), four studies included costs of care for both mothers and children over the entire cost accrual period, and one study only included costs of treating the child. In most studies (12, 75%) costs were estimated over the course of the pregnancy and during the birth, with five of these also including postpartum costs three to twelve months after the birth. Two studies only included postpartum costs, one included antenatal care costs only, and one study only included costs associated with the birth and one year postpartum (Table 1).

Incremental costs per case associated with gestational diabetes ranged from €263 (South Korean insurance claims registry data, mothers' costs only during pregnancy and birth[36]) to

€13,680 (US modelling study, mothers and neonatal care costs during pregnancy and birth [32]). Average incremental pregnancy and postpartum cost estimates were higher for US versus non-US studies (€4,607 versus €1,444). Higher average costs were also reported for studies that modelled resource use and costs compared to those based on an analysis of cross-sectional data (€4,890 versus €2,040), and for studies that included mother and child costs compared to those that only included costs of treating mothers (€3,388 versus €2,048).

## Obesity

Of the 13 studies that reported costs associated with overweight or obese mothers, 11 examined costs of care for mothers or children during pregnancy, birth or postpartum. One study reported higher costs of care for children of overweight or obese mothers over the first 18 years of life, and one study compared the costs of treating minor complications during pregnancy, such as heartburn and carpal tunnel syndrome in overweight/obese and normal weight women (Table 2). All studies adopted a payer perspective, with seven limiting the analysis to maternal care costs, three focusing on infant care costs, and three including cost elements from both mothers and children.

Estimates of the average incremental costs associated with pregnancy and birth in overweight or obese mothers ranged from €191 (UK cohort study estimating costs of mothers care during birth[26]) to €16,046 (US modelling study estimating costs of maternal and neonatal care during pregnancy and birth[32]). As with the evidence base for diabetes, average incremental costs of obesity were higher for studies set in the US (€6,867 versus €768), modelling studies (€16,046 versus €2,459), and studies that included costs of neonatal care as well as maternal care (€8,964 versus €1,612).

## Mental health problems

A total of eight studies reported incremental costs associated with mental health problems (Table 3). Two of these modelled direct and indirect care costs over the lifetime of mothers and children, reporting significant intergenerational care costs associated with maternal mental health problems.[21, 22] One study reported higher annual postnatal costs of care for women with depression, and another reported higher costs for children of mothers with depression, excluding costs associated with pregnancy and birth.[20, 35]

Among the four studies that examined costs during pregnancy, birth or the immediate postpartum period the estimated incremental costs of poor maternal mental health ranged from €452 to €794.[28, 53] All of these studies adopted a payer perspective with postpartum follow up ranging from four weeks to one year. Only one of these studeis was set in the US (average incremental costs per case were €576), and all estimates were obtained from analysis of cross-sectional data on resource use and costs.

## Other maternal health problems

Seven studies reported incremental costs associated with hypertensive disorders (four studies), nausea and vomiting (two studies), epilepsy (two studies) or exposure to intimate partner violence (1 study, Table 4).

Hypertensive disorders included both confirmed pre-eclampsia (one Irish and one US study[25, 56]) and hypertension during pregnancy (three studies set in the US[30, 37, 56]), with all hypertension studies examining different costs (mothers only, children only or mother and children combined). Incremental costs per case for hypertension ranged from €2,860 to €8,595 (Table 4).

**Table 2. Incremental costs of overweight/obesity.**

| Study (Setting) | Study design (n) | Accrual period for costs | Perspective-Costs included | Results [absolute cost increase (percentage increase)] |
|---|---|---|---|---|
| Kuhle 2018[46] (Canada) | Longitudinal (34,820) | First 18 years of life | Payer-Children only | Having an overweight (€226, %NR) or obese (€1,160, %NR) mother was associated with higher costs of care over the first 18 years of a child's life |
| Solmi 2018[26] (UK) | Cross-sectional (7,091) | Birth only | Payer-Mothers only | Higher unadjusted cost among mothers who were overweight (BMI 25–29: €45, 3%), obese level 1 (BMI 30–34: €165, 11%), and obese level 2/3 (BMI ≥35: €254, 17%) |
| Law 2015[37] (US) | Cross-sectional (137,040) | First 3 months of life | Payer-Children only | Higher unadjusted cost of new-born care among mothers with obesity (€3,028, 38%) |
| Morgan 2015[47] (UK) | Cross-sectional (609) | 12 months postpartum | Payer-Children only | Average annual care costs were higher in infants of overweight (€79, 4%) and obese (€1,392, 72%) mothers |
| Whiteman 2015 [38] (US) | Cross-sectional (576,843) | Birth and 12 months postpartum | Payer-Mothers and children | Being overweight/obese was associated with higher average costs of maternal and infant care (€774, 11%) |
| Caldas 2015[48] (US) | Cross-sectional (167) | During pregnancy and birth | Payer-Mothers and children | Obesity associated with higher maternity and childcare costs (€10,071, 37%, including hospital and physician costs) |
| Lenoir-Wijnkoop 2015[32] (US) | Modelling (N/A) | During pregnancy and birth | Payer-Mothers and neonatal care | Overweight or obese mothers (€16,046, %NR) were associated with higher costs of care. |
| Law 2015[30] (US) | Cross-sectional (322,141) | During pregnancy and 3 months postpartum | Payer-Mothers only | Higher unadjusted cost of maternal care for those with obesity (€4,802, 35%) |
| Denison 2014[31] (UK) | Cross-sectional (120,673) | During pregnancy and birth | Payer-Mothers only | Being overweight (€269, 8%), obese (€711, 21%) and severely obese (€1,299, 39%) were associated with higher costs compared to normal weight mothers |
| Morgan 2014[49] (UK) | Cross-sectional (484) | During pregnancy and 2 months postpartum | Payer-Mothers only | Significantly higher costs associated with obese women (€1,500, 33%), higher mean costs for overweight mothers (€894, 20%) were not statistically significant |
| Watson 2013[50] (Australia) | Cross-sectional (36,331) | During pregnancy to 90 days postpartum | Payer-Mothers only | Higher mean costs for overweight (€337, 6%), obese I (€609, 11%), obese II (€959, 17%), and obese III (€1,216, 22%) mothers, compared to normal weight mothers |
| Trasande 2009[51] (US) | Cross-sectional (232,315) | Per pregnancy-related episode of care | Payer-Mothers only | A secondary diagnosis of obesity was associated with higher average costs across all pregnancy-related hospitalisations (€2,643, %NR), and also after adjusting for the rate of CS (€1,998, %NR) |
| Denison 2009[52] (UK) | Cross-sectional (651) | 10–12 weeks gestation to birth | Payer-Mothers only | Higher costs associated with treating minor complications for overweight (€3, 14%) and obese (€49, 215%) mothers |

NR not reported; BMI body mass index; CS caesarean section

Two studies compared costs of moderate or severe nausea or vomiting during pregnancy to those with mild symptoms using different methods (modelling versus primary data analysis, payer versus societal perspective, US versus Canadian costs).[24, 57] Estimates of the incremental costs of maternal care for severe nausea and vomiting ranged from €191 to €454, Table 4).

Two studies reported incremental costs of either maternal care only (one study, €6,033[30]) and infant care only (one study, €1,694[37]) for births complicated by maternal epilepsy. Both of these studies were based on US data (Table 4).

A single study reported incremental costs associated with intimate partner violence (IPV). [29] This found that births among women who experience IPV are associated with increased costs from longer hospital stays, greater prevalence of clinical conditions such as sexually transmitted diseases and depression, and poorer infant outcomes, such as preterm birth (Table 4).

Box plots presented in Fig 2 illustrate the spread of the incremental cost estimates associated with each of the included morbidities. This figure only includes data from studies reporting costs from the start of pregnancy to a maximum of 12 months postpartum, and excludes

**Table 3. Incremental costs of mental health problems.**

| Study (Setting) | Study design (n) | Accrual period for costs | Perspective-Costs included | Results [absolute cost increase (percentage increase)] |
|---|---|---|---|---|
| Moore Simas 2019 [35] (US) | Cross-sectional (135,678) | 24 months postpartum | Payer- Children only | Depression associated with incremental costs of €2,019 (12%) over the first 2 years of the childs life |
| Chojenta 2018[54] (Australia) | Cross-sectional (3,684) | Pregnancy and 1 year postpartum | Payer-Mothers only | History of poor mental health is associated with an average increase of €507 (11%) in costs per birth |
| Ammerman 2016 [20] (US) | Cross-sectional (20,440) | Annual postpartum costs— not pregnancy or birth costs | Societal-Mothers only | Depression associated with greater probability of incurring expenses (OR 1.51) as well as higher expenditure for those treated, resulting in an average incremental cost of €1,564 (55%) per woman |
| Bauer 2016[21] (UK) | Modelling (N/A) | Lifetime of mothers and children | Societal-Mothers and children | Estimated net present value of per person lifetime costs of depression (€92,642, %NR) and anxiety (€42,586, %NR). |
| Bauer 2015[22] (UK) | Modelling (N/A) | Lifetime of children | Societal-Children only | Higher costs associated with exposure to maternal depression for both the public sector (€4,010, %NR) and for society (€10,838, %NR), which included productivity and HRQoL losses. |
| Dagher 2012[55] (US) | Cross-sectional (638) | From birth to 11 weeks postpartum | Payer-Mothers only | Depression associated with higher mean expenditure per woman (€576, 186%) |
| Petrou 2002[28] (UK) | Cross-sectional (206) | Birth to 18 weeks postpartum | Payer (Public sector) -Mothers and children | Having postnatal depression was associated with a non-significant increase of €794 (19%) in costs |
| Roberts 2001[53] (Canada) | Cross-sectional (873) | 0–4 weeks postpartum | Payer-Mothers and children | Depression was associated with higher costs (€452, 105%) at 4 weeks postpartum |

OR odds ratio; N/A not applicable; NR not reported

**Table 4. Incremental costs of hypertensive disorders, nausea and vomiting, epilepsy, and intimate partner violence.**

| Study (Setting) | Study design (n) | Accrual period for costs | Perspective-Costs included | Results [absolute cost increase (percentage increase)] |
|---|---|---|---|---|
| **Hypertensive disorders** | | | | |
| Hao 2019[56] (US) | Cross-sectional (2,136) | 20 weeks gestation to 6 weeks postpartum (mothers) or 12 months (infants) | Payer-Mothers and neonatal care | Preeclampsia (€22,360, 217%) and hypertension (€8,595, 83%) were both associated with higher costs of care |
| Fox 2017[25] (Ireland) | Cross-sectional (233) | 15 weeks gestation to 12 months postpartum | Payer-Mothers and neonatal care | Women with preeclampsia were associated with higher costs of care (€2,860, 114%) |
| Law 2015[30] (US) | Cross-sectional (322,141) | During pregnancy and 3 months postpartum | Payer-Mothers only | Higher unadjusted cost of maternal care for those with hypertension (€5,382, 40%) |
| Law 2015[37] (US) | Cross-sectional (137,040) | First 3 months of life | Payer-Children only | Higher unadjusted cost of new-born care among mothers with hypertension (€8,174, 112%) |
| **Nausea and vomiting** | | | | |
| Piwko 2013[57] (US) | Modelling (N/A) | During pregnancy (per woman) | Payer-Mothers only | Higher treatment costs for moderate (€15, 43%) and severe NVP (€191, 568%) compared to mild NVP |
| Piwko 2007[24] (Canada) | Cross-sectional (139) | During pregnancy (per woman-week) | Societal-Mothers only | Moderate (€194, 169%) and severe (€454, 395%) NVP was associated with higher costs per women week compared with mild NVP |
| **Epilepsy** | | | | |
| Law 2015[30] (US) | Cross-sectional (322,141) | During pregnancy and 3 months postpartum | Payer-Mothers only | Higher unadjusted cost of maternal care for those with epilepsy (€6,033, 44%) |
| Law 2015[37] (US) | Cross-sectional (137,040) | First 3 months of life | Payer-Children only | Higher unadjusted cost of new-born care among mothers with epilepsy (€1,694, 21%) |
| **Intimate partner violence** | | | | |
| Mogos 2016 [29] (US) | Cross-sectional (32,658,259) | Birth only | Payer-Mothers and neonatal care | Intimate partner violence was associated with higher costs of birth-related discharges (€1,410, 33%) |

N/A not applicable; NVP nausea and vomiting in pregnancy

studies that only reported long-term postpartum costs (that did not involve pregnancy/birth costs), or those that were restricted to costs of infant/childcare only. Fig 2A shows the absolute incremental costs per case (in 2018 Euro) and Fig 2B shows the proportional increase compared to those without the condition. In absolute terms, obesity was associated with the largest spread of estimates of incremental costs, followed by hypertensive disorders and diabetes. The largest proportional change in costs was observed in severe nausea and vomiting, because while the amounts are small, they are multiples of the costs of care for those who experience only mild symptoms. Obesity, in contrast, is associated with significant absolute cost differences that translate into relatively smaller proportional increases.

## Discussion

Our result show that relatively common health problems experienced by women during pregnancy and postpartum impose a substantial economic burden on health systems and society. The conditions for which most evidence is available are gestational diabetes, obesity, and depression, all of which are associated with increases in the average treatment costs per

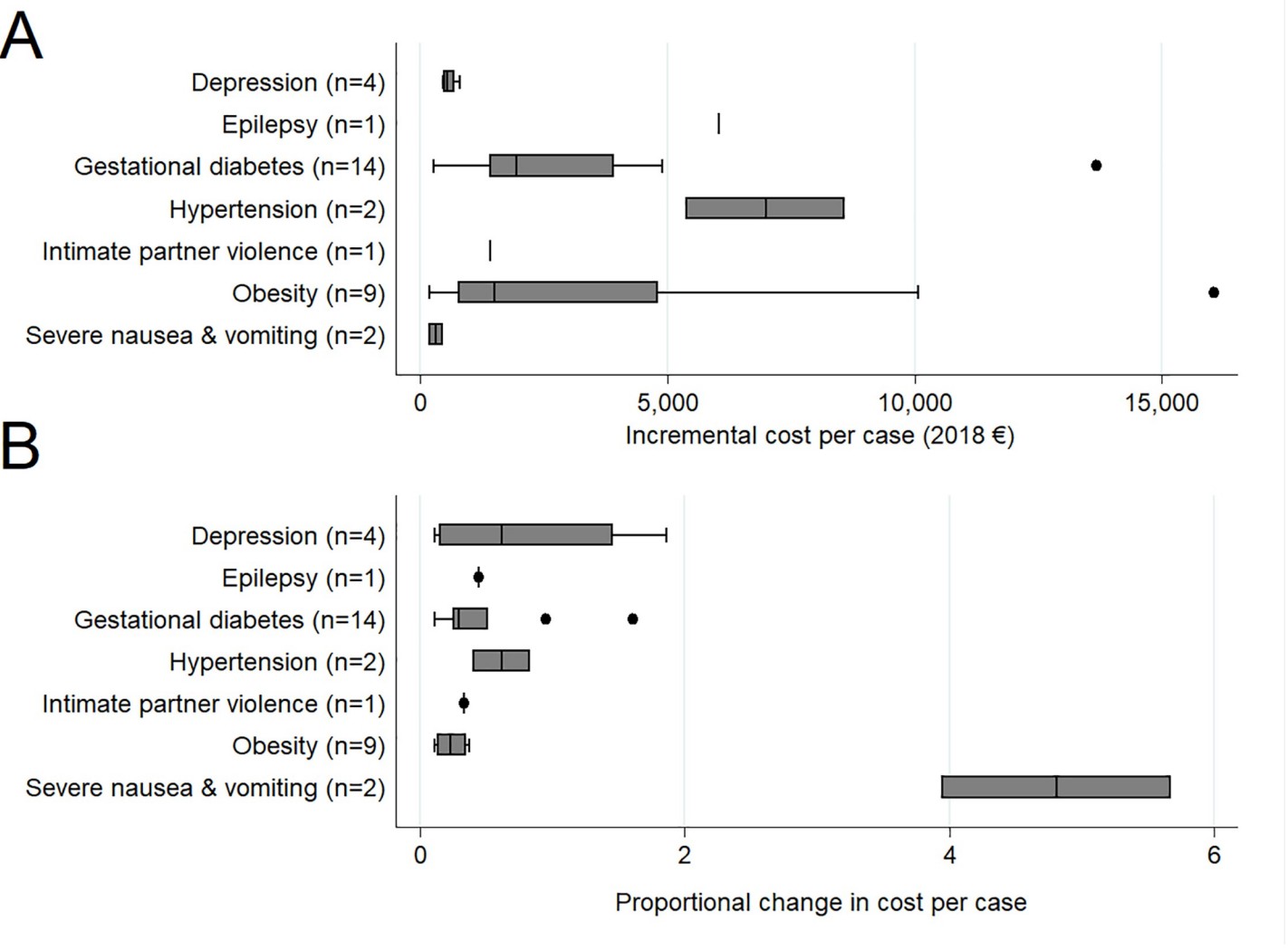

**Fig 2.** Absolute (A) and proportional (B) increase in costs from pregnancy to 12 months postpartum.

women, as well as subsequent increases in the cost of caring for children of mothers who experienced these health problems. There is also evidence to indicate that hypertension, epilepsy, and severe nausea and vomiting are associated with increased costs, as is exposure to intimate partner violence during pregnancy. No studies were identified that examined the economic burden of other common maternal morbidities, such as incontinence, pelvic girdle or back pain, exhaustion, or sexual health problems.

### Methodological issues

The evidence base identified in this review was characterised by a high level of clincial and methodological heterogeneity in both the study population and methods used, as reflected by the range of incremental cost estimates obtained for each morbidity (Fig 2). The most common analytical approach involved partitioning cross-sectional data based on whether or not women had a particular morbidity, and then comparing costs between these groups. While the ability to aggregate cohort and registry data like this has the benefit of facilitating robust analysis of even relatively rare events, a lack of detailed information on case detection and clinical management tends to limit our interpretation of the incremental cost estimates, and the extent to which the economic burden could be lessened through improved clinical management of these conditions.

Modelling studies, which bring together the best available information from a range of sources to simulate costs in a hypothetical population of women, have the potential to overcome some of these limitations by giving researchers the ability to compare specific differences in care pathways for women with or without a given health problem, and distinguish costs associated with clinical intervention from those associated with managing the consequences of a lack of intervention. However, the external validity of these types of studies is dependent on the quality of the data used and any assumptions made by researchers. An example of the challenges of modelling is provided by two excluded studies examining the costs associated with discontinuation of antidepressant therapy during pregnancy that were conducted at the same time in Canada.[58, 59] Both studies sought to examine the expected increase in resource use due to the rise in depression relapse rates that had been observed elsewhere after discontinuing treatment, one by modelling and the other by analysis of registry data. While the modelling study reported significant cost savings associated with maintaining treatment, the registry study found that when the costs of the antidepressants themselves were excluded, there was no significant difference between the two groups. This apparent contradiction stemmed from an inability to adjust for differences in casemix and disease severity between groups that chose to discontinue therapy or not, highlighting the difficulty in reliably modelling changes in clinical outcomes and costs when faced with selective uptake within the target population. This may be of relevance to our review, as we identified large differences in the incremental cost estimates obtained from modelling studies compared to studies that analysed primary data. However, the underlying causes of these differences were not identifiable.

The studies included in this review were carried out in eleven different countries. This contributes to the diversity observed in the results, since setting can significantly influence the magnitude of any incremental cost differences associated with maternal health problems. Although we used purchasing power parities to account for differences in price levels between countries, it is unlikely to fully adjust for differences in funding and reimbursement arrangements or the overall structure of healthcare in different settings. Only for diabetes and obesity were sufficient studies identified to compare costs by setting, and this found that average cost estimates were higher for US versus non-US studies. However, there was quite a degree of methodological heterogeneity within each of these groups across other characteristics, such as

cost accrual period and study design. One way to adjust for differences in the baseline costs of care in different health systems is to express the change in terms of a proportional increase rather than an absolute increase. When doing this we found the proportional increases in the costs associated with obesity and diabetes were relatively consistent, in contrast with depression which had a much wider spread of estimates relative to the baseline cost of care in each country.

The quality appraisal process revealed weaknesses in the evidence base in relation to the inclusion of costs that fell outside the payer perspective. Eight studies adopted a broader societal perspective and five of these included productivity losses associated with time off work. Given that most maternal morbidity is primarily experienced by women in the community, adopting a payer perspective that is limited to the public health system or health insurance company risks masking the significant economic burden that falls on women and families in dealing with the consequences of health problems that are often underreported and undertreated.[7, 8, 10] Ultimately, the most informative result from a cost of illness study is the net present value of the incremental costs of a particular health problem over as long a time frame as is needed to capture all the consequences of that problem, in everyone who is affected by it. This, of course, presents challenges, and there are inevitable trade-offs between the scope of the analysis and its degree of external validity. Only one included study estimated incremental costs over the lifetime of mothers and children from a societal perspective, focussing specifically on perinatal anxiety and depression.[21] This found that maternal mental health problems impose a significant economic burden on UK society (£6.6 billion for each annual birth cohort). Perhaps more importantly, these type of studies reveal valuable information about the nature of these costs, such as the significant economic burden of maternal mental health problems on children as a result of poorer physical and mental health, decreased quality of life, and reduced career prospects in later life. It also found that only about a fifth of the total cost fell on the public health and social care system, with the majority being incurred by the individuals themselves or society. While other studies examined some of these issues, such as the intergenerational costs of obesity and diabetes, no comparable estimates are available for other morbidities.

Broader analysis of how women and families cope with the economic burden of maternal ill health has only been examined in studies focussing on severe obstetric complications during birth for women in developing countries.[60–64] Our search identified five studies that explored how households adapted to the economic shocks associated with maternal ill health, and how it affected families' consumption and borrowing, and the distribution of these impacts across different socioeconomic groups in society. While these studies were ineligible due to the type of maternal morbidity (acute obstetric complications), they highlight the absence of comparable information about women with less severe maternal morbidity during and after pregnancy.

Quality appraisal also highlighted limitations in regard to the handling of uncertainty about costs estimates, and a lack of sensitivity analysis examining key assumptions used in studies (see S2 Table).

## Other reviews

We are unaware of any previous systematic review of cost-of-illness studies on non-acute maternal morbidity before and after pregnancy. One review of cost-of-illness studies on reproductive, maternal, newborn, and child health was identified, which included a number of studies examining complications such as preterm birth, non-exclusive breastfeeding, or sexually transmitted diseases.[65] Similar to our review, this study reported significant heterogeneity

due to methodological differences in the design, costing approach, perspective, and time horizon of included studies. A 2006 WHO review of the costs of maternal-newborn illness and mortality identified one peer-reviewed publication on the cost components of emergency obstetric care, but did not report any data on non-acute maternal health problems of interest in our review.[66] Finally, a review of the costs of pregnancy in the US identified 26 studies reporting costs of pregnancy-related complications.[67] These primarily related to costs associated with pre-term birth and low birthweight, and did not include any of the studies in our review.

## Limitations

Identification of relevant studies was challenging given the broad scope of the review and the difficulty in devising a search strategy to limit the results to cost of illness studies. To overcome this a two-stage approach was taken, which involved first running a broad search with high expected sensitivity, followed by a more focussed search that had high expected specificity. Full details of the search strategy are included as supplementary material. Despite these precautions, it is unlikely that all relevant studies were included in this review, particularly studies in which incremental costs were reported as a secondary outcome. As far as we are aware this review is the first to draw together the available evidence on the economic burden of these types of common health problems associated with pregnancy and childbirth, so although it is unlikely to be exhaustive, it represents the most comprehensive attempt at identifying this information so far available in the literature.

## Conclusion

Our findings indicate that maternal health problems experienced by women during pregnancy and postpartum (as opposed to acute complications of labour or birth or severe acute adverse maternal outcomes) are associated with significant costs to health systems and society. However, there is considerable methodological heterogeneity in study designs, cost accrual periods and choice of perspective, which is reflected in the wide range of estimates for the incremental costs associated with each maternal health problem. Most research to date has focused on gestational diabetes, obesity and depression, but important gaps remain in the evidence base for the economic burden of some common morbidities associated with pregnancy and childbirth, such as incontinence, pelvic girdle or back pain, exhaustion, or sexual health problems. More research is needed to examine the economic burden of these maternal health problems, and future research should adopt consistent methodological approaches to ensure comparability of results.

## Supporting information

**S1 Checklist. PRISMA checklist.**
(DOCX)

**S1 Table. Search strategy.**
(PDF)

**S2 Table. Data extraction table for all included studies.**
(PDF)

**S3 Table. Results of quality appraisal.**
(PDF)

## Author Contributions

**Conceptualization:** Patrick S. Moran, Stephanie Brown, Cecily Begley, Deirdre Daly.

**Data curation:** Patrick S. Moran, Francesca Wuytack.

**Formal analysis:** Patrick S. Moran, Francesca Wuytack, Cecily Begley, Deirdre Daly.

**Funding acquisition:** Cecily Begley.

**Methodology:** Patrick S. Moran, Francesca Wuytack, Charles Normand, Deirdre Daly.

**Supervision:** Michael Turner, Charles Normand, Cecily Begley.

**Validation:** Patrick S. Moran, Francesca Wuytack, Stephanie Brown, Cecily Begley, Deirdre Daly.

**Writing – original draft:** Patrick S. Moran.

**Writing – review & editing:** Patrick S. Moran, Francesca Wuytack, Michael Turner, Charles Normand, Stephanie Brown, Cecily Begley, Deirdre Daly.

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
