## [Decision Letter · Decision Letter 0]

5 Nov 2019

PONE-D-19-18054

Economic burden of maternal morbidity – a systematic review of cost-of-illness studies

PLOS ONE

Dear Dr Moran,

Thank you for submitting your manuscript to PLOS ONE. After careful consideration, we feel that it has merit but does not fully meet PLOS ONE’s publication criteria as it currently stands. Therefore, we invite you to submit a revised version of the manuscript that addresses the points raised during the review process.

As well as the reviewer comments I have some minor points that I feel should be addressed: 

Abstract…………. Conclusion: Perhaps a sentence suggesting what could be done to improve the evidence base?

Line 91 Among the benefits of COI analysis is that it provides a detailed description of how much society is spending on a particular health problem, and therefore how much would be saved if it could be eradicated completely……………………. But what if the health problem can never be irradiated or treated effectively? Maternal birth injury for example cannot be eradicated completely, it is present to some degree with all vaginal and caesarean births?

It is now 6 months since the search………… could there be other studies now published that should be included? I suggest you update the search

Please spell out all abbreviations in full in the first instance: OECD?

One issue with combining classes of obesity is that level I obesity will likely be less costly compared to level III………….. perhaps a limitation you should comment on?

Line 223 what was the entire accrual period?

We would appreciate receiving your revised manuscript by the 25th November. To enhance the reproducibility of your results, we recommend that if applicable you deposit your laboratory protocols in protocols.io, where a protocol can be assigned its own identifier (DOI) such that it can be cited independently in the future. For instructions see: http://journals.plos.org/plosone/s/submission-guidelines#loc-laboratory-protocols

We look forward to receiving your revised manuscript.

Kind regards,

Diane Farrar

Academic Editor

PLOS ONE

Journal Requirements:

Reviewers' comments:

Reviewer's Responses to Questions

**Comments to the Author**

1. Is the manuscript technically sound, and do the data support the conclusions?

Reviewer #1: Partly

Reviewer #2: Yes

Reviewer #3: Yes

2. Has the statistical analysis been performed appropriately and rigorously? 

Reviewer #1: No

Reviewer #2: Yes

Reviewer #3: Yes

3. Have the authors made all data underlying the findings in their manuscript fully available?

Reviewer #1: Yes

Reviewer #2: Yes

Reviewer #3: Yes

4. Is the manuscript presented in an intelligible fashion and written in standard English?

Reviewer #1: Yes

Reviewer #2: Yes

Reviewer #3: Yes

5. Review Comments to the Author

Reviewer #1: Manuscript is a systematic review trying to assess the economic burden of maternal morbidity and explore gaps for evidence.

- Please include statistical means of describing "high level of heterogeneity" and display heterogeneity statistics accordingly. You can stratify studies by level of heterogeneity and examine subgroup meta-analysis.

- Maternal morbidity is known to vary across different countries in the world. Poor countries in Africa have higher maternal morbidity and mortality compared to developed nations, and economic burden also correlates to such difference. Any attempt to describe the economic impact and variation of global maternal morbidity will be helpful.

Reviewer #2: I found this to be an interesting and useful paper. I only have a minor comment which is related to the significant variation in the incremental cost estimates associated with each maternal health problem. The authors suggest this variation is due to differences in design, accrual periods and choice of perspective, but, do they detect patterns in the types of studies and contexts that explains that variability?

Reviewer #3: This well-written and interesting manuscript makes a critically important contribution to the literature. The economic burden of maternal morbidity has been overlooked in the research literature, but is needed for the epidemiology literature is clear that among certain populations, complex and multiple morbidity disadvantage mothers caring for themselves and their families.

This manuscript highlights the limitations of the research to date. The economic burden beyond that to the health care payers, and the immediate childbearing year needs a more expanded perspective. This manuscript points out where the gaps are in understanding more fully the costs to society of highly prevalent distressing postpartum problems. Economics drive health care policy. Thus, the research implications of this manuscript are important as policy makers need to understand how to optimize resources for women's health clinical and public health services.

6. PLOS authors have the option to publish the peer review history of their article (what does this mean?). If published, this will include your full peer review and any attached files.

Reviewer #1: No

Reviewer #2: No

Reviewer #3: Yes: Joan Rosen Bloch, PhD, CRNP, FAAN

---

## [Author Response · Author response to Decision Letter 0]

25 Nov 2019

The authors would like to thank the Editor and reviewers for their comments, which are addressed individually below.

Editor comments:

1. Abstract…………. Conclusion: Perhaps a sentence suggesting what could be done to improve the evidence base?

We agree and have added the following sentence to the abstract “More research is needed to examine the economic burden of a range of common maternal health problems, and future research should adopt consistent methodological approaches to ensure comparability of results.”, along with the following text in the conclusion in the body of the paper “More research is needed to examine the economic burden of these maternal health problems, and future research should adopt consistent methodological approaches to ensure comparability of results.”

2. Line 91 Among the benefits of COI analysis is that it provides a detailed description of how much society is spending on a particular health problem, and therefore how much would be saved if it could be eradicated completely……………………. But what if the health problem can never be irradiated or treated effectively? Maternal birth injury for example cannot be eradicated completely, it is present to some degree with all vaginal and caesarean births?

While complete eradication is not a realistic prospect for most diseases, analysis of the economic burden they place on health systems and society can provide important information to help inform priority setting, as well as laying the foundations for assessment of the cost effectiveness of measures designed to prevent or treat them. We endeavoured to present both the strengths and weaknesses of the type of analysis we carried out as clearly and transparently as possible, to aid readers interpreting the results. This is addressed in paragraphs 2 and 3 of the introduction. 

3. It is now 6 months since the search………… could there be other studies now published that should be included? I suggest you update the search

The search has been updated, resulting in the inclusion of two additional studies. All relevant sections of the manuscript and supplementary material file have been updated accordingly. 

4. Please spell out all abbreviations in full in the first instance: OECD?

Done (OECD, BMI).

5. One issue with combining classes of obesity is that level I obesity will likely be less costly compared to level III………….. perhaps a limitation you should comment on?

We agree, and have highlighted this limitation by adding the following sentence to the paragraph explaining how we grouped some subpopulations to facilitate cross study comparisons – “While this was necessary to facilitate comparisons with other studies that used similar groupings, a limitaton of this is that it does not permit comparisons across different obesity categories (e.g. class I versus class III obesity).”

6. Line 223 what was the entire accrual period?

The cost accrual period (i.e. length of time over which costs were counted) in this study (Anderberg et al., Scand J Public Health, 2012. 40(4): p. 385-90) was between 10 and 14 years. To eliminate any potential confusion in the use of accrual in this context (accrual of costs as opposed to recruitment) we have changed the term to “cost accrual period” instead of “accrual period” throughout the manuscript.

Reviewer #1 comments: 

1. Manuscript is a systematic review trying to assess the economic burden of maternal morbidity and explore gaps for evidence. Please include statistical means of describing "high level of heterogeneity" and display heterogeneity statistics accordingly. You can stratify studies by level of heterogeneity and examine subgroup meta-analysis.

The type of heterogeneity being referred to here is not statistical heterogeneity, where the incremental costs reported in seemingly similar studies differ by more than than one would expect due to random error, but rather clinical and methodological heterogeneity, due to differences in the populations included in different studies examining the same morbidity, or differences in the methodology used to to estimate incremental costs. While this helps explain the wide range of estimates that we found, it also means that it is not appropriate to combine these results to produce a pooled estimate. We have clarified this by specifying that it is clinical and methodological heterogeneity that it being referred to in the revised manuscript. 

The suggestion to use subgroup analysis is something that we considered at length, as this can potentially be used to investigate differences in outcomes between studies that use different methodological approaches, but given the limited number of studies within each disease area and the degree of diversity in populations and methods, opportunities for subgroup analysis of this type were limited. We did, however, conduct subgroups analysis of the impact of geographical setting, by comparing US versus non-US studies where possible, and we also compared the result of studies that were based on analysis of primary data compared to those in which incremental costs were modelled by combining data from multiple sources in a secondary data analysis.

2. Maternal morbidity is known to vary across different countries in the world. Poor countries in Africa have higher maternal morbidity and mortality compared to developed nations, and economic burden also correlates to such difference. Any attempt to describe the economic impact and variation of global maternal morbidity will be helpful.

As mentioned above, we did seek to examine differences in cost in different settings. However given the relatively few included studies for each morbidity, the only subgroup analysis the data would facilitate involved comparing costs in US versus non-US studies. This showed that the costs of care in the US tend to be higher than elsewhere. 

In this review we specifically sought to examine the costs of maternal health problems that are generally less severe but occur over a longer period of time than the type of acute obstetric emergencies that are the focus of much of the existing literature, such as depression, incontinence, etc. and we did not include studies that examined the economic impact of maternal mortality on households and on wider society.

Reviewer #2 comment: 

1. I found this to be an interesting and useful paper. I only have a minor comment which is related to the significant variation in the incremental cost estimates associated with each maternal health problem. The authors suggest this variation is due to differences in design, accrual periods and choice of perspective, but, do they detect patterns in the types of studies and contexts that explains that variability?

One of the main findings is the high level of variability in the evidence base in terms of study populations, methodological approaches, and incremental costs. The reviewer is of course right in pointing out that in these circumstances the aim should be to detect patterns that help make sense of this diversity and provide insight into the factors associated with higher or lower costs. However, while we did look at changes in cost by setting (US versus non-US was the only comparison that we could make for this) and by whether studies were based on analysis of primary or secondary data (regression analyses versus modelling studies), there was insufficient data to carry out any other meaningful subgroup analyses within each disease area. Indeed such was the level of diversity in the results that some analyses that involved relatively short cost accrual periods or a narrower perspective reported higher incremental costs than those that included a wider range of costs over a longer time period. In summary, therefore, we did seek to understand the discrepancies in incremental costs observed in the review, but apart from limited analyses by setting (US vs non-US and methodology (primary vs secondary data analyses) we were not able to detect specific factors that explained this variability. Though not a subgroup analyses per se, we also report incremental costs within a subset of included studies that all spanned the hospital admission for the birth episode and included maternal care costs, and presented the spread of results graphically, as both the absolute difference in costs, as well as the proportional change in costs compared to those without a given morbidity (Figure 2).

Reviewer #3 comments: 

2. This well-written and interesting manuscript makes a critically important contribution to the literature. The economic burden of maternal morbidity has been overlooked in the research literature, but is needed for the epidemiology literature is clear that among certain populations, complex and multiple morbidity disadvantage mothers caring for themselves and their families. This manuscript highlights the limitations of the research to date. The economic burden beyond that to the health care payers, and the immediate childbearing year needs a more expanded perspective. This manuscript points out where the gaps are in understanding more fully the costs to society of highly prevalent distressing postpartum problems. Economics drive health care policy. Thus, the research implications of this manuscript are important as policy makers need to understand how to optimize resources for women's health clinical and public health services.

We would like to thank the reviewer for her comments.

---

## [Decision Letter · Decision Letter 1]

18 Dec 2019

Economic burden of maternal morbidity – a systematic review of cost-of-illness studies

PONE-D-19-18054R1

Dear Dr. Moran,

We are pleased to inform you that your manuscript has been judged scientifically suitable for publication and will be formally accepted for publication once it complies with all outstanding technical requirements.

With kind regards,

Diane Farrar

Academic Editor

PLOS ONE

Additional Editor Comments (optional):

Reviewers' comments:

Reviewer's Responses to Questions

**Comments to the Author**

1. If the authors have adequately addressed your comments raised in a previous round of review and you feel that this manuscript is now acceptable for publication, you may indicate that here to bypass the “Comments to the Author” section, enter your conflict of interest statement in the “Confidential to Editor” section, and submit your "Accept" recommendation.

Reviewer #1: All comments have been addressed

Reviewer #2: All comments have been addressed

2. Is the manuscript technically sound, and do the data support the conclusions?

Reviewer #1: Yes

Reviewer #2: Yes

3. Has the statistical analysis been performed appropriately and rigorously? 

Reviewer #1: Yes

Reviewer #2: Yes

4. Have the authors made all data underlying the findings in their manuscript fully available?

Reviewer #1: Yes

Reviewer #2: (No Response)

5. Is the manuscript presented in an intelligible fashion and written in standard English?

Reviewer #1: Yes

Reviewer #2: Yes

6. Review Comments to the Author

Reviewer #1: (No Response)

Reviewer #2: (No Response)

7. PLOS authors have the option to publish the peer review history of their article (what does this mean?). If published, this will include your full peer review and any attached files.

Reviewer #1: No

Reviewer #2: No

---

## [Editor Report · Acceptance letter]

9 Jan 2020

PONE-D-19-18054R1 

Economic burden of maternal morbidity – a systematic review of cost-of-illness studies 

Dear Dr. Moran:

I am pleased to inform you that your manuscript has been deemed suitable for publication in PLOS ONE. Congratulations! Your manuscript is now with our production department. 

With kind regards,

on behalf of

Dr. Diane Farrar 

Academic Editor

PLOS ONE